# Development and validation of the Self-Efficacy in Addressing Menstrual Needs Scale (SAMNS-26) in Bangladeshi schools: A measure of girls' menstrual care confidence

**Erin C. Hunter**[1,2]*, **Sarah M. Murray**[3], **Farhana Sultana**[4,5], **Mahbub-Ul Alam**[4], **Supta Sarker**[4], **Mahbubur Rahman**[4], **Nazrin Akter**[4], **Moshammot Mobashara**[4], **Marufa Momata**[4], **Peter J. Winch**[2]

**1** Sydney School of Public Health, Faculty of Medicine and Health, The University of Sydney, New South Wales, Australia, **2** Department of International Health, Johns Hopkins Bloomberg School of Public Health, Baltimore, Maryland, United States of America, **3** Department of Mental Health, Johns Hopkins Bloomberg School of Public Health, Baltimore, Maryland, United States of America, **4** Environmental Interventions Unit, Infectious Diseases Division, icddr,b, Dhaka, Bangladesh, **5** School of Public Health and Preventive Medicine, Monash University, Melbourne, Victoria, Australia

* erin.hunter@sydney.edu.au

**Data Availability Statement:** To align with the informed consent provided by study participants, approval is needed for other researchers to access

## Abstract

### Objective

Qualitative studies have described girls' varying levels of confidence in managing their menstruation, with greater confidence hypothesized to positively impact health, education, and social participation outcomes. Yet, measurement of this and other psychosocial components of adolescent girls' menstrual experiences has been weak in global health research, in part due to a dearth of appropriate psychometric tools. We describe the development and validation of the Self-Efficacy in Addressing Menstrual Needs Scale (SAMNS-26).

### Methods

We conducted nine focus group discussions with girls in schools in rural and urban Bangladesh to identify tasks involved in menstrual self-care. This informed our creation of an initial pool of 50 items, which were reviewed by menstrual health experts and refined through 21 cognitive interviews with schoolgirls. Using a self-administered survey, we administered 34 refined items plus additional validation measures to a random sample of 381 post-menarcheal girls (ages 9–17) and retested a subsample of 42 girls two weeks later. We examined the measure's dimensionality using exploratory factor analysis and assessed internal consistency, temporal stability, and construct validity.

### Results

Exploratory factor analysis suggested a 26-item scale comprising three correlated subscales: the 17-item Menstrual Hygiene Preparation and Maintenance (α = 0.86), the 5-item Menstrual Pain Management (α = 0.87), and the 4-item Executing Stigmatized Tasks (α =

the quantitative data. Data are available from the icddr,b institutional data repository for researchers upon approval of a Data Licensing Application & Agreement. For more information, see https://www.icddrb.org/component/content/article/10003-datapolicies/1893-data-policies. Request for icddr,b research data should be addressed to Ms. Armana Ahmed, Head, Research Administration at aahmed@icddrb.org. Making the qualitative data (audio recordings) publicly available would compromise the confidentiality we promised to study participants, as girls could be identifiable by their voices and stories. We have provided a high-level summary of qualitative data generated during focus group discussions as supporting info with the published paper.

**Funding:** The 'Piloting MHM interventions among urban and rural schools in Bangladesh' study (the "main study") was funded by the Bill and Melinda Gates Foundation (OPP1140650) (https://www.gatesfoundation.org/) to FS. icddr,b acknowledges with gratitude the commitment of BMGF to its research efforts. icddr,b is also grateful to the Governments of Bangladesh, Canada, Sweden, and the UK for providing core/unrestricted support. Additional support for data collection towards the self-efficacy sub-study was provided through a Dissertation Enhancement Award to ECH from the Center for Qualitative Studies in Health and Medicine of Johns Hopkins Bloomberg School of Public Health (https://www.jhsph.edu/departments/health-behavior-and-society/research-and-centers/center-for-qualitative-studies-in-health-and-medicine/). The funders had no role in study design, data collection and analysis, decision to publish, or preparation of the manuscript.

**Competing interests:** The authors have declared that no competing interests exist.

0.77). Sub-scales exhibited good temporal stability. SAMNS-26 scores correlated negatively with measures of anxiety, and girls who preferred to stay at home during their periods had lower SAMNS-26 scores than those who did not.

## Conclusion

The SAMNS-26 provides a reliable measure of a schoolgirl's confidence in her capabilities to address her menstrual needs. There is initial evidence to support the measure's construct validity in the Bangladesh context as indicated by its relationships with other factors in its theorized nomological network. The tool enables incorporation of self-efficacy into multivariate models for exploring the relationships among antecedents to menstrual experiences and hypothesized impacts on health, wellbeing, and education attainment. Further testing of the tool is recommended to strengthen evidence of its validity in additional contexts.

## Introduction

### Background

Schoolgirls in low- and middle-income country (LMIC) settings cope with a variety of challenges in meeting their needs regarding menstruation [1, 2]. While typically provided little education about menstruation, adolescent girls contend with menstrual stigma, inadequate sanitation and disposal facilities, limited access to reliable menstrual materials, and poor social support for menstrual issues [1–3]. Over the past decade, there has been increasing advocacy to implement school-based programs to ensure girls can adequately and comfortably address their needs related to menstruation in the school environment—both because it is critical for girls' human rights to dignity and reproductive health, and also because it is essential for achieving gender equality in education [4–8].

School-based menstrual health intervention studies have conventionally aimed to improve menstrual knowledge and management practices with a view to reducing school absenteeism, but they have often stopped short of rigorously assessing other important components of girls' menstrual experiences [9, 10]. The relationship between menstrual practices and school attendance is an indirect one—likely mediated by menstrual factors (e.g., dysmenorrhea, menorrhagia, etc.) and psychosocial factors such as agency and confidence, shame, distress, perceptions of one's environment, and perceptions of one's own menstrual practices [1, 10]. Measurement of these important psychosocial factors in menstrual health research and program evaluations has been weak, in part due to a dearth of appropriate tools specific to this domain [11]. Validated measures for the assessment of psychosocial components of adolescent girls' menstrual experiences are critical for robust evaluation studies to establish strong evidence for school-based interventions that effectively improve menstrual experiences.

### Theoretical foundation

Prior qualitative studies in LMICs have described women and girls' varying levels of confidence in managing their menstruation, with greater confidence hypothesized to positively impact health, education, and social participation outcomes [1]. Menstrual care confidence is an important psychosocial factor to assess in the menstrual health and hygiene domain because beliefs in one's capabilities are known to influence how individuals feel, think, motivate themselves, and act [12]. Guided by Bandura's self-efficacy theory [12], we conceptualize

menstrual care confidence beliefs as "self-efficacy" and define the construct as a girl's *beliefs in her capabilities to carry out the tasks required to address her menstrual needs.* An individual's self-efficacy beliefs are constructed through processing information drawn from their previous enactive experiences, vicarious experience through observing others similar to themselves, verbal persuasion (e.g., encouragement), and their own physiological and affective state [12].

In menstrual health intervention studies, assessing self-efficacy beliefs may provide a more comprehensive picture of girls' lived experiences than objective assessments of the presence of sanitation facilities in schools or availability of menstrual materials [1]. Self-efficacy theory suggests that even among those who share the same physical and economic environments, girls with lower self-efficacy in addressing their menstrual needs will perceive opportunities and constraints differently than those who have greater self-efficacy. Those with low self-efficacy in addressing their menstrual needs may be likely to experience greater stress and anxiety while attending to their menstruation in the face of challenges. Such individuals are theoretically more likely to avoid challenging situations altogether [12]—such as attending school during menstruation.

## Measurement of self-efficacy

Bandura's self-efficacy theory conceptualizes the self-efficacy construct as comprising three dimensions: strength, level/magnitude, and generality—which has implications for its measurement [12]. The strength dimension is assessed by having participants indicate on a Likert-type response scale how sure they are that they can perform a particular task [13]. Along the second dimension, the level/magnitude of self-efficacy beliefs refers to the level of difficulty of task demands at which individuals feel they are capable of succeeding [12]. Self-efficacy beliefs also differ along a dimension of generality—or the "degree to which the expectation is generalized across situations" [12, 14]. When measuring self-efficacy beliefs, it is thus important to measure the strength of an individual's perceived capability across varying degrees of challenge or impediments to successful performance in a variety of situations [12, 13]. Doing so necessitates multi-item measures. The purpose of the present study was to develop and refine a pool of potential items for a *Self-Efficacy in Addressing Menstrual Needs Scale* and formally test the items to reduce the pool and preliminarily assess the measure's psychometric properties with schoolgirls in Bangladesh.

## Methods

### Study setting

Data collection took place in eight schools in Bangladesh—four in urban Dhaka, and four in rural Manikganj District—that were participating in the larger study *Piloting menstrual hygiene management interventions among urban and rural schools in Bangladesh* (henceforth referred to as the "main study"). The main study comprised a formative research phase in four of the schools (two urban, two rural) to inform the development of an intervention package, followed by a six-month piloting period in four other schools (two urban, two rural) to evaluate the intervention. The intervention aimed to promote supportive school environments for menstruating girls. Intervention components included provision of schoolteacher-led puberty and menstruation education, improved waste disposal facilities, and distribution of menstrual materials and menstrual cycle tracking calendars (among other activities). Details about the study context, school selection and characteristics, and the main study's intervention activities and evaluation methods have been presented separately [15, 16].

The work described in this paper was a self-efficacy sub-study commenced halfway through the main study (S1 Fig), after the formative research phase but immediately prior to

implementation of a baseline survey in intervention schools. We first leveraged the main study's baseline survey conducted in the four intervention schools to assess feasibility of our tool's instructions and response options. We then conducted qualitative research to inform the content of the scale items in the four schools that had earlier participated in the main study's formative research phase. Lastly, we leveraged the main study's endline survey in the intervention schools to collect quantitative data for the self-efficacy scale development.

The Ethical Review Committee of icddr,b approved the study protocol (PR15115). The Dhaka Zonal Office, Directorate of Secondary and Higher Education; the Dhaka Divisional Office, Directorate of Primary Education; and School Management Committees provided permission to conduct research in the schools. School leadership further provided approval to conduct study activities on school property, primarily during school snack breaks to reduce disruption. All participants assented to participate and had written consent from a parent or schoolteacher as their guardian (*in loco parentis*).

## Overview of research design

Fig 1 summarizes our four-stage process model for developing the *Self-Efficacy in Addressing Menstrual Needs Scale* between April 2017 and April 2018. In Phase 1, we designed the questionnaire format and created an initial pool of draft items. In Phase 2, we assessed the content validity of the item pool through an expert validation exercise. In Phase 3, we iteratively pretested and refined the item pool and conducted final field piloting of the tool. In Phase 4, we assessed the psychometric properties of the tool through testing items on a survey of randomly selected schoolgirls in urban and rural Bangladesh.

## Phase 1: Creation of questionnaire format and initial item pool

**Design and feasibility testing of questionnaire format, instructions, and response options.**   Guided by existing measures of self-efficacy in other domains [13, 17], we developed a generic format for items that directs a respondent to indicate the strength of her confidence in her capability to perform a particular task involved in addressing her menstrual needs. We chose an 11-point Likert-type response option to allow enough variation in responses as recommended by Bandura for self-efficacy scales [13].

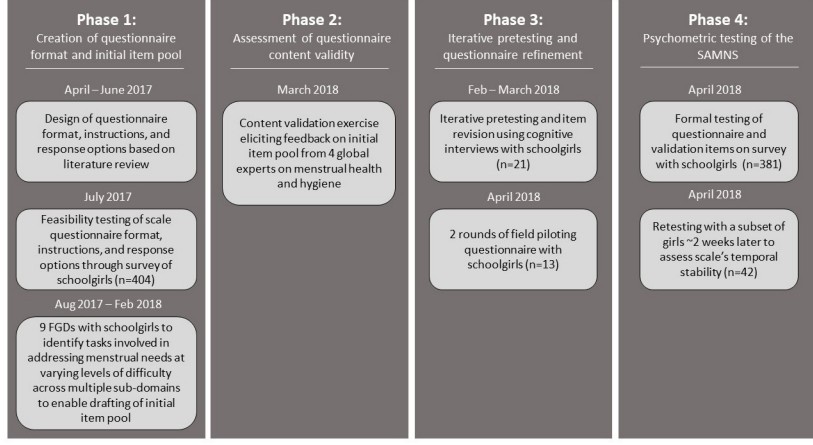

**Fig 1.  Process of development and testing of the Self-Efficacy in Addressing Menstrual Needs Scale in 8 urban and rural schools in Bangladesh 2017–2018.**

We trained 14 female professional health survey enumerators to pilot 10 test items on the main study's baseline survey in the four intervention schools to qualitatively assess acceptability and feasibility of our proposed self-efficacy questionnaire format, instructions, and response options. During training, survey enumerators recommended using response options 0 to 100 (in intervals of 10) because of schoolgirls' familiarity with the 0–100 scale used in academic grading. Among several options for visual cues to aid in understanding response options, survey enumerators recommended an "X" over the "0" response option, and a check/tick that increased in size over subsequent response options. Word labels anchored either end of the scale where 0 = "No, I absolutely cannot do it" and 100 = "Yes, I am absolutely sure I can definitely do it" (Fig 2).

We devised a "pen test" to check participants' comprehension of the instructions before collecting data [13]. A survey enumerator would place a pen on the table near to a participant and ask, "*How confident are you that you can reach the pen (while remaining in your seat)*?" with the expectation that the response should be at or near 100. The enumerator would proceed to move the pen progressively farther away and ask the same question. If responses did not move in a reasonable manner along the response options, then the enumerator would provide further explanation of the instructions to clarify misunderstandings. In July 2017, we observed the main study's baseline survey implementation with 527 randomly selected girls in Classes 5–10 (404 of whom had reached menarche and so participated in the feasibility testing of the self-efficacy questionnaire's instructions and response options) and debriefed with the survey enumerator team about their experiences conducting the feasibility testing.

**Development of item pool.** Having confirmed the acceptability and feasibility of the intended format for our scale, we compiled a list of actions involved in menstrual self-care to write a comprehensive pool of potential self-efficacy items. To this end, we first reviewed global literature on girls' menstrual experiences to identify categories of tasks related to addressing menstrual needs: obtaining menstrual materials; using, changing, disposing, and/or washing menstrual materials; keeping bodies clean during menstruation; reducing menstrual pain or discomfort; seeking support, help or advice related to menstruation; and coping with stress or anxiety related to menstruation.

Then, between August 2017 and February 2018, we conducted nine focus group discussions (FGDs) with schoolgirls from Classes 4–10 to explore how they understood the meaning of

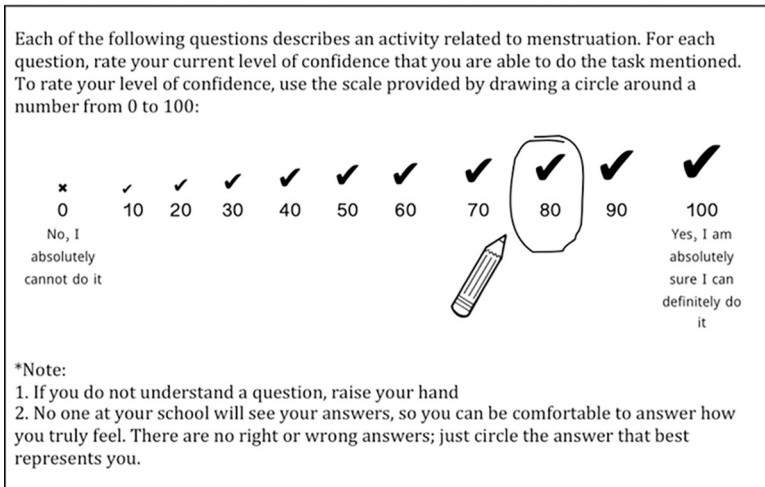

**Fig 2. Questionnaire instructions and response options developed for the Self-Efficacy in Addressing Menstrual Needs Scale, tested in Bangladesh, 2017–2018.**

*menstrual management* and to identify various tasks they viewed as part of meeting menstrual needs across the identified categories. We used participatory activities based on vignettes (S2 Fig) to facilitate discussion due to the research topic being socially proscribed and the potential for participants to feel shy discussing menstruation in a group.

To recruit and select participants, we worked with female schoolteachers to explain the purpose of the research in classrooms of female students and compile a list of post-menarcheal girls by class level and approximate time since menarche. Schoolgirls voluntarily self-identified their eligibility in the absence of males. We then instructed eligible participants on how to request written consent from parents to participate in the study and requested students return signed consent forms to us the following day. We later purposively selected participants from this master list of eligible girls for each data collection episode according to class level and length of time since menarche. Focus group discussions were convened with privacy in empty classrooms. They were conducted in Bengali (*Bangla*) and lasted 84 minutes on average. The study team took detailed field notes and debriefed immediately after each FGD to expand field notes, chart findings, write analytic memos, and to make sampling decisions for subsequent FGDs. We charted findings from each FGD by completing a table with columns for 1) menstrual care tasks identified across various categories, 2) situations that would make those tasks more difficult, and 3) situations that would make the tasks less difficult. FGDs were audio recorded so we could check any gaps in field notes, but full transcriptions were not produced for the analyses presented in this paper. Data collection continued until we stopped identifying new tasks and felt confident we could write a sufficiently large pool of potential scale items [18, 19]. Using these data and our review of the literature, we wrote an initial pool of 50 items, including items at varying levels of difficulty to avoid a ceiling effect in the final scale.

## Phase 2: Assessment of questionnaire content validity

We invited five global experts on menstrual health and hygiene to provide feedback in March 2018 via online survey to assess content validity of the initial pool of 50 draft items. Experts provided qualitative feedback regarding our conceptualization of the construct and then rated the relevance of each item from 1–4 where 1 = *not relevant*, 2 = *unable to assess relevance without item revision*, 3 = *relevant but needs minor alteration*, and 4 = *very relevant and succinct*. They also provided comments on the clarity of each item and made suggestions for revision. We revised items with average ratings of less than 4 according to experts' suggestions.

## Phase 3: Iterative pretesting and questionnaire refinement

**Pretesting using cognitive interviews.**   In February and March 2018, we conducted cognitive interviews to pretest and refine the pool of draft items. Cognitive interviewing enables survey developers to examine whether participants can easily understand questions and whether they interpret items the way they were intended—which aids in identifying concrete suggestions for improving items and response options [20]. We conducted 21 cognitive interviews in iterative rounds (2–6 interviews per round) with girls in Classes 5–10 in the same urban and rural schools where we had conducted FGDs. This iterative approach enabled us to identify and address issues with items and test revised versions until we achieved a good match between each item's intent and participants' interpretations [21]. We purposively selected girls from the master list of eligible students who had not previously participated in FGDs and according to time since menarche to ensure a range of familiarity with menstrual care.

Cognitive interviews were conducted privately in empty classrooms. Study team members used a field guide that contained all draft items being tested, each with suggested probing questions and blank space for field notes. We encouraged participants to express their thoughts

aloud as they interpreted the items, searched their memory to formulate an answer, and chose how to select a response from the response scale [22, 23]. Study team members also probed concurrently to encourage participants to verbalize their thought processes and to explore any confusion that arose or uncertainty over word meanings or instructions [24]. We applied both proactive (anticipated and spontaneous) and reactive (conditional and emergent) probes [24, 25]. Interviews were conducted in Bengali, typically lasted 1–1.5 hours, and were audio recorded for verification purposes but not transcribed. Led by the first author, the study team convened immediately after interviews to debrief, combine field notes from all interviews, and revise items to improve clarity. The revised list of items served as the field guide for the following round. Items were dropped before final field piloting if they were not comprehensible during cognitive interviewing despite multiple efforts at revision or if response frequency distribution charts showed very little variation in quantitative responses.

**Field piloting the scale questionnaire.** We conducted a final field pilot in two rounds in April 2018 with a total of 13 girls selected by convenience from Classes 8–10 in an urban school to determine how long it took them to complete the revised pool of 35 items and if they could do so without assistance. The iterative approach provided an opportunity to make adjustments between rounds if necessary. We asked retrospective probing questions after girls completed the tool to check whether their interpretation of items matched our intention [20, 24]. We dropped one item following field piloting, leaving a total of 34 items to be formally tested in Phase 4.

## Phase 4: Psychometric testing

**Study sample and survey procedures.** In April 2018, we leveraged the main study's endline survey in the four intervention schools (two urban, two rural) to collect data for assessing the dimensionality, reliability, and validity of the 34-item menstrual care self-efficacy tool and to determine which, if any, items should be dropped. The main study's endline survey was administered to 528 girls randomly selected from class rosters. After completing the survey, only girls who had experienced menstruation were directed to two study team members who oversaw the self-efficacy sub-study survey.

Study team members provided brief verbal instructions to participants (one-to-one or in small groups) and conducted the "pen test" for comprehension of the instructions before asking each participant to complete the pen-and-paper questionnaire on her own. A random sub-sample of 42 girls completed the questionnaire again approximately two weeks later to test temporal stability. Data from the hardcopy questionnaires were entered into a spreadsheet by one study team member and checked by one or two others to ensure fidelity.

**Survey measures.** The self-efficacy sub-study survey had two sections: the first comprised 34 self-efficacy items for formal testing (S1 File). The second section comprised three additional measures for use in the assessment of the self-efficacy tool's construct validity: the Bengali (*Bangla*) versions [26] of the Beck Self-Concept Inventory for Youth (BSCI-Y) and Beck Anxiety Inventory for Youth (BAI-Y) [27], and our Bengali translation of the Social Self-Efficacy Scale [28].

Although often conflated, self-efficacy is a construct related to yet distinct from self-concept or self-worth [12]. Therefore, we expected not to see a strong correlation between the self-efficacy tool and the BSCI-Y. Informed by self-efficacy theory, we anticipated that self-efficacy scores would be negatively correlated with measures of anxiety. However, since the BAI-Y is not a specific measure of menstrual-related state anxiety, we hypothesized that correlations would be low to moderate. We included social self-efficacy as a validation measure because some of our tool's items involve interacting with and obtaining assistance from others to

address menstrual needs. Although we hypothesized there would be a positive correlation between scores on our tool and Social Self-Efficacy Scale scores, we wanted to ensure that our finalized measure was not so highly correlated with the construct of social self-efficacy that it would not provide any additional utility.

Additional items from the main study's endline survey were used for validity testing. The items "During my last period, I felt anxious at school because of my menstruation" and "During my last period, I felt comfortable at school" served as indicators of anxious arousal more specifically related to menstrual experiences than the BAI-Y. The item "I prefer to stay at home during my period" was used to validate whether girls with lower self-efficacy scores tended to desire avoidance of contexts that make addressing menstrual needs challenging. Participants' responses to these items on a 6-point Likert-type response scale of 1 = strongly agree to 6 = strongly disagree were dichotomized as "agree" or "disagree" for this analysis.

We hypothesized that girls' self-efficacy scores would positively correlate with time (months) since menarche, calculated from two items on the main study's survey: current age and age when menstruation first began.

(See S2 File for additional details of the survey measures.)

**Analytic strategy for psychometric testing.** We applied factor analysis methods to empirically explore how many unobservable constructs underlie the set of 34 formally tested items. This was to provide an understanding of whether *self-efficacy in addressing menstrual needs* should best be measured in our sample as one broad construct or rather through multiple more specific constructs (or "sub-scales") [29]. In preparation for the exploratory factor analysis (EFA), we assessed all items for missing values, examined charts of item response frequency distributions, and performed tests of multivariate normality using the STATA command "omninorm" [30]. We then examined the item correlation matrix and used Bartlett's test of sphericity [31] to ensure the data showed mild collinearity. Lastly, sampling adequacy was assessed using the Kaiser-Meyer-Olkin (KMO) measure (with *a priori* minimum acceptable threshold set at 0.50) [32].

We applied three methods to determine the number of factors to extract in EFA. We first conducted a principal components analysis (PCA) to identify how many components had eigenvalues greater than one. We then examined a scree plot and performed a parallel analysis test. We tested whether the extracted factors were correlated above Tabachnick and Fidell's threshold of 0.32 [33], and subsequently used oblimin oblique normalized rotation. We made an *a priori* decision to consider dropping items that failed to load at least 0.30 on any factor during EFA, or if any items loaded highly on more than one factor [34].

Internal consistency was assessed for each sub-scale by calculating Cronbach's α and by examining the average and individual interitem correlations [35, 36]. Items that contributed poorly to internal consistency were considered for elimination. We calculated sub-scale scores for each observation by calculating the sum of responses for each retained item divided by the total number of items in the sub-scale. Temporal stability was assessed by retesting a subsample of girls two weeks after the first administration of the survey and calculating the concordance correlation coefficient for scale scores at the two time points. Bland-Altman plots were also examined to assess test-retest agreement in individuals' scores. The time interval between test and retest was chosen to minimize the chance of substantial real changes in self-efficacy beliefs.

We examined initial evidence for construct validity in our sample by assessing relationships between self-efficacy scale scores and other validation measures. For validation measures treated as continuous, we created scatter plots to visualize the relationships and calculated Spearman correlations. For dichotomous variables, we performed Wilcoxon rank-sum tests to

test for differences in distributions of self-efficacy scores between groups. All analyses were conducted using Stata, version 14.2 (StataCorp LP, College Station, TX).

## Results

### Feasibility testing of questionnaire format, instructions, and response options

The 404 girls who participated in the feasibility testing of the tool's intended format, instructions, and response options ranged in age from 10 to 18. Our observation of the feasibility testing and debriefing with the survey enumerator team indicated that enumerators found it easy to explain the instructions for the self-efficacy items and the item format made sense to the schoolgirls. Enumerators found that the "pen test" enabled them to quickly identify and clarify girls' misunderstandings about the response options.

### Item pool development

A total of 51 girls ages 11–16 years participated in the nine FGDs (socio-demographic information available in S1 Table). Five FGDs (3 urban, 2 rural) involved girls who were two years or less post-menarche while four FGDs (2 urban, 2 rural) involved girls who had been menstruating for more than two years. A summary of example menstrual care tasks discussed by our FGD participants, along with the conditions under which those tasks might be made more difficult or less difficult to enact is available in S2 Table. This chart guided the development of our initial pool of 50 draft scale items.

We had not included "being prepared for menstruation" as an *a priori* category of menstrual care tasks in the participatory activities, yet its salience to schoolgirls was identified in FGDs. We therefore also drafted items within this category (e.g., items regarding a girl's confidence in her capability to track her menstrual cycle or estimate approximately when her next period will begin—which involve body literacy).

### Expert validation exercise

Four out of the five invited experts participated in the content validation exercise. The overall average relevance rating for items was 3.52 out of 4. Experts did not suggest the inclusion of additional categories of tasks, therefore indicating our initial item pool appeared to have good content validity. No items were dropped solely based on expert feedback.

### Cognitive interviews

Cognitive interview participants were ages 11–16 years (socio-demographic information available in S1 Table). Thirteen girls (9 urban, 4 rural) were 6–24 months post-menarche, and 8 girls (6 urban, 2 rural) had reached menarche more than two years prior.

Most item revisions performed after each round of cognitive interviews concerned word choice and sentence structure. However, most items in the category of "keeping bodies clean during menstruation" were dropped due to lack of variation in responses during cognitive interviews. All girls interviewed felt fully confident they could wash their genital area and body as needed during their menstrual periods; therefore, such items would not contribute meaningfully to a scale that is intended to differentiate among girls with varying self-efficacy beliefs in this context. We also dropped items referring to the washing and drying of menstrual materials, because girls who only used disposable materials were unsure how to respond. We retained items referring to disposal of menstrual materials since even those who use cloth

eventually dispose of it. Revisions based on cognitive interviewing reduced the pool of draft items from 50 to 35.

The items were intended to be relevant for girls regardless of the type of menstrual materials they use, but cognitive interviews revealed the difficulty in selecting a generic term to encompass any type of material used for absorbing or collecting menstrual blood. Therefore, each time the term "menstrual material" appeared in an item, we added a list of the most commonly used materials in this context so girls could more easily understand what was being referenced (e.g., pad, cloth, cotton, tissue, etc.). Table 1 displays two selected examples of item revisions based on cognitive interview findings.

During cognitive interviews we noticed that girls were better able to focus on, process, and respond to the items when they could read them themselves. We therefore shifted from developing an interviewer-administered self-efficacy scale to a self-administered one, which would reduce administration time.

## Field piloting

The average time to complete the 35 items during final field piloting was 15.5 minutes with a range of 10 to 22 minutes. Girls requested little clarification about item meanings during questionnaire administration, and retrospective probing indicated that most girls had understood the items as intended. After piloting, we further clarified wording in items about pain management and removed an item that asked about confidence to reduce abdominal pain "by a medium amount" because of its similarity to other items and the need to reduce administration time.

**Table 1. Example item revisions based on findings from cognitive interviews with schoolgirls during the development of the Self-Efficacy in Addressing Menstrual Needs Scale in Bangladesh, 2018.**

| Original item | Problems identified | Revised item(s) |
|---|---|---|
| *"How confident are you that you can do the things necessary to manage your period when you are visiting someone else's house where both males and females are present?"* | Meaning of "manage your period" was not readily understood<br>Answers were very different depending on whether girls imagined "someone else's house" referring to that of a family member versus that of a stranger | *"Imagine you are at a relative's home and it becomes necessary to change the menstrual material you're wearing (such as: pad, cloth, tissue, cotton, etc.). How confident are you that you can change it there?"*<br>*"How confident are you that you can change your menstrual material (such as: pad, cloth, tissue, cotton, etc.) if it becomes necessary while you're at a female friend's house (without returning to your own home)?"* |
| *"How confident are you that you can track your menstrual cycle on the calendar?"* | Meaning of "track your menstrual cycle" was not readily understood<br>Girls reported low confidence not due to their incapacity to track their cycles per se (the specific task of interest), but rather because families commonly have only one calendar. It would be embarrassing to mark certain days on the calendar where others could see it. The *specific* method of tracking cycle length is not particularly important, so item should be revised to be relevant for girls who use any method. | *"How confident are you that you can count/keep track of your period days?"* |

## Survey sample characteristics

Of 382 girls who indicated on the main study's endline survey they had reached menarche, 381 completed the self-efficacy sub-study survey. No items were missing values among the 381 completed surveys. S3 Table displays the characteristics of participants disaggregated by urban and rural. The mean age was 14 years (SD = 1.5), and mean age at menarche was approximately 12 years (SD = 1.2). Regarding type of menstrual materials used, 61.7% of girls reported using disposable pads while in their homes, and 58.3% reported using pieces of cloth and/or reusable pads to absorb menses (participants could select multiple options). While away from home, 70.3% reported using disposable pads and 41.2% reported using pieces of cloth and/or reusable pads. The majority (66.4%) of girls reported experiencing menstrual pain during their last period, with 41.1% of those respondents describing their pain as "severe." For those girls experiencing menstrual pain, most (55.3%) reported a one-day duration of pain.

## Item reduction and scale dimensionality

We present item response mean, standard deviation, skew, and kurtosis for all 34 self-efficacy items in S4 Table. We display the items we dropped during the psychometric testing and our rationales in S5 Table. We dropped four self-efficacy items because they showed very little variation in responses, and thus would not contribute meaningfully to a scale [35]. We dropped two additional items because they did not correlate with any other items at a level of at least 0.30 [37]. The Bartlett's test of sphericity [31] for the retained 28 items was significant ($p < 0.001$) and the Kaiser-Meyer-Olkin (KMO) measure of sampling adequacy was 0.88, indicating our data were appropriate for factor analysis [32, 37, 38].

We employed iterated principal factors as our method of factor extraction due to the data not being multivariate normal. Performing a PCA on the retained 28 items showed seven components with eigenvalues greater than one. The scree plot showed an elbow at three factors, and the parallel analysis test suggested a 3-factor solution. In the subsequent 3-factor EFA, the item that assessed a girl's confidence in her capability to remove a bloodstain from her clothes while at school failed to load at least 0.30 on any factor, so we dropped this item and re-ran the EFA. Upon assessing internal consistency of the three factors, we determined that dropping an item that asked about a girl's confidence in her capabilities to use hot fomentation to reduce menstrual pain would increase the second factor's internal consistency. The 26-item, 3-factor final model accounted for 43.5% of the total variance. Table 2 displays the rotated factor loadings (pattern matrix) and uniqueness for this final model, while Table 3 displays the structure matrix consisting of correlations between items and the rotated factors. Table 4 displays the correlations among the three factors.

All 17 items loading on factor 1, which we labeled *Menstrual Hygiene Preparation and Maintenance (MHPM)*, measured girls' confidence in their capabilities to accomplish tasks related to obtaining, using, and changing menstrual materials in a variety of contexts; seeking assistance for menstrual hygiene when needed; and anticipating days of bleeding (factor loadings: 0.31–0.60). All five items loading on factor 2, labeled *Menstrual Pain Management (MPM)*, measured confidence in mitigating menstrual pain (factor loadings: 0.59–0.87). The four items loading on factor 3, labeled *Executing Stigmatized Tasks (EST)* measured girls' confidence in their capabilities to accomplish menstrual care tasks that are heavily affected by menstrual stigma—particularly involving the risk of disclosing menstrual status in the presence of males (factor loadings: 0.41–0.95). We refer to the finalized tool (S3 and S4 Files) as the Self-Efficacy in Addressing Menstrual Needs Scale (SAMNS-26).

**Table 2. Factor loadings based on exploratory factor analysis with oblimin oblique normalized rotation of 26 retained Self-Efficacy in Addressing Menstrual Needs items (n = 381).**

| Abbreviated item | Factors | | | Unique-ness |
|---|---|---|---|---|
| | 1 (MHPM) | 2 (MPM) | 3 (EST) | |
| SE2: Imagine you are at a relative's home and it becomes necessary to change the menstrual material you're wearing. . . you can change it there | **0.31** | 0.04 | 0.19 | 0.80 |
| SE3: . . . you can change your menstrual material at school if it becomes necessary | **0.53** | -0.13 | -0.07 | 0.79 |
| SE5: . . . you can change your menstrual material if it becomes necessary while you're at a female friend's house | **0.53** | -0.05 | 0.14 | 0.64 |
| SE7: . . . you can properly use a menstrual material so that menstrual blood does not stain your clothing while participating in school sports | **0.49** | 0.06 | -0.04 | 0.75 |
| SE8: . . . you can participate in your normal daily activities during your period without worry that your menstrual material will become displaced | **0.50** | 0.00 | 0.10 | 0.69 |
| SE9: . . .you can walk quickly during your period without your menstrual material becoming displaced | **0.60** | -0.08 | 0.02 | 0.66 |
| SE11: If the menstrual material that you use most often is not available. . .you can use another type of menstrual material instead | **0.38** | 0.03 | 0.10 | 0.79 |
| SE12: . . . you can lie down during your period without bloodstaining the bed sheet during the night | **0.55** | 0.09 | -0.12 | 0.70 |
| SE13: . . . you are able to **try** to reduce abdominal pain during your period if it becomes necessary | 0.27 | **0.59** | -0.19 | 0.52 |
| SE14: . . . you can reduce abdominal pain during your period | 0.04 | **0.82** | -0.02 | 0.31 |
| SE17: . . . you can dispose of a used menstrual material if a male person is nearby | 0.16 | 0.03 | **0.42** | 0.72 |
| SE19: . . . you are at school and your period starts but you have not brought your own menstrual material. . .you are able to obtain a menstrual material somehow in that moment to meet your need while still at school | **0.52** | 0.02 | 0.02 | 0.71 |
| SE20: . . .if necessary, you're able to ask a female friend for a menstrual material | **0.38** | 0.08 | 0.02 | 0.81 |
| SE21: . . . you can take help from a female teacher if you face a menstrual-related problem at school | **0.56** | 0.04 | -0.01 | 0.67 |
| SE22: . . . you can ask *aya** for help regarding your menstruation if a male teacher is nearby | 0.21 | 0.11 | **0.41** | 0.65 |
| SE23: Imagine pads are available at school. . .you can go ask for a pad by yourself when you need it, without the help of friends | **0.49** | 0.01 | 0.21 | 0.60 |
| SE24: Imagine you have the money to purchase a pad. . .you can ask a male seller at a pharmacy for a pad | 0.10 | -0.05 | **0.67** | 0.49 |
| SE25: Imagine you have the money to purchase a pad. . .you can ask a pharmacy seller for a pad when there are male persons around | -0.03 | 0.02 | **0.95** | 0.12 |
| SE26: . . . you can roughly predict when your period is about to start | **0.38** | 0.20 | 0.06 | 0.72 |
| SE27: . . . you are able to prevent bloodstaining your clothing even while traveling a long distance during your period | **0.41** | 0.10 | 0.04 | 0.77 |
| SE28: . . . if Sir/Madam asks a question in class, you can stand up to answer during your period without worry that you have bloodstained your clothing | **0.53** | 0.00 | 0.10 | 0.65 |
| SE29: . . . when you need menstrual materials you can obtain them even if a trusted female is not available at home | **0.34** | 0.06 | 0.19 | 0.76 |
| SE30: . . . you can count/keep track of your period days | **0.34** | 0.19 | 0.10 | 0.75 |
| SE32: . . . you can usually reduce your abdominal pain by a **small amount** | 0.17 | **0.66** | -0.05 | 0.46 |
| SE33: . . . you can usually reduce **most** of your abdominal pain | -0.14 | **0.87** | 0.12 | 0.27 |
| SE34: . . . you can usually reduce your abdominal pain **completely** | -0.10 | **0.79** | 0.09 | 0.40 |

Notes: Factor loadings over 0.30 in bold; *MHPM* Menstrual hygiene preparation and maintenance, *MPM* Menstrual pain management, *EST* Executing stigmatized tasks
*aya* refers to the women who worked in the study schools as janitors and caretakers. Schoolgirls typically had friendly relationships with these women, and if the school provided any menstrual pads for emergencies, it was typically the *aya* who could give one to a student.

## SAMNS-26 scores

The mean MHPM sub-scale score for all 381 participants was 74.4 (SD = 16.6) with a range of 23.5 to 100. The mean MPM sub-scale score was 64.7 (SD = 26.5) with a range of 0 to 100. The mean EST sub-scale score was 45.5 (SD = 28.6) with a range of 0 to 100. Table 5 displays mean scores disaggregated by geography and use of disposable pads while away from home during the most recent menstrual period. Girls in urban schools tended to have slightly higher scores than girls in rural schools across all three sub-scales, although the differences were not significant. Girls who ever used disposable pads while away from their home during their most

**Table 3. (Structure matrix) correlations between Self-Efficacy in Addressing Menstrual Needs items and the oblimin oblique normalized rotated common factors (n = 381).**

| Abbreviated item | Factors | | |
|---|---|---|---|
| | 1 | 2 | 3 |
| SE2: Imagine you are at a relative's home and it becomes necessary to change the menstrual material you're wearing. . . you can change it there | 0.42 | 0.21 | 0.35 |
| SE3: . . . you can change your menstrual material at school if it becomes necessary | 0.44 | 0.07 | 0.16 |
| SE5: . . . you can change your menstrual material if it becomes necessary while you're at a female friend's house | 0.58 | 0.21 | 0.40 |
| SE7: . . . you can properly use a menstrual material so that menstrual blood does not stain your clothing while participating in school sports | 0.49 | 0.25 | 0.22 |
| SE8: . . . you can participate in your normal daily activities during your period without worry that your menstrual material will become displaced | 0.55 | 0.23 | 0.35 |
| SE9: . . .you can walk quickly during your period without your menstrual material becoming displaced | 0.58 | 0.17 | 0.31 |
| SE11: If the menstrual material that you use most often is not available. . .you can use another type of menstrual material instead | 0.45 | 0.21 | 0.30 |
| SE12: . . . you can lie down during your period without bloodstaining the bed sheet during the night | 0.53 | 0.29 | 0.18 |
| SE13: . . . you are able to **try** to reduce abdominal pain during your period if it becomes necessary | 0.42 | 0.65 | 0.09 |
| SE14: . . . you can reduce abdominal pain during your period | 0.37 | 0.83 | 0.20 |
| SE17: . . . you can dispose of a used menstrual material if a male person is nearby | 0.38 | 0.20 | 0.51 |
| SE19: . . . you are at school and your period starts but you have not brought your own menstrual material. . .you are able to obtain a menstrual material somehow in that moment to meet your need while still at school | 0.54 | 0.24 | 0.29 |
| SE20: . . .if necessary, you're able to ask a female friend for a menstrual material | 0.43 | 0.25 | 0.24 |
| SE21: . . . you can take help from a female teacher if you face a menstrual-related problem at school | 0.57 | 0.27 | 0.28 |
| SE22: . . . you can ask *aya**\*** for help regarding your menstruation if a male teacher is nearby | 0.46 | 0.30 | 0.54 |
| SE23: Imagine pads are available at school. . .you can go ask for a pad by yourself when you need it, without the help of friends | 0.60 | 0.27 | 0.46 |
| SE24: Imagine you have the money to purchase a pad. . .you can ask a male seller at a pharmacy for a pad | 0.42 | 0.15 | 0.71 |
| SE25: Imagine you have the money to purchase a pad. . .you can ask a pharmacy seller for a pad when there are male persons around | 0.45 | 0.24 | 0.94 |
| SE26: . . . you can roughly predict when your period is about to start | 0.49 | 0.37 | 0.30 |
| SE27: . . . you are able to prevent bloodstaining your clothing even while traveling a long distance during your period | 0.47 | 0.28 | 0.27 |
| SE28: . . . if Sir/Madam asks a question in class, you can stand up to answer during your period without worry that you have bloodstained your clothing | 0.58 | 0.25 | 0.37 |
| SE29: . . . when you need menstrual materials you can obtain them even if a trusted female is not available at home | 0.46 | 0.24 | 0.38 |
| SE30: . . . you can count/keep track of your period days | 0.47 | 0.35 | 0.31 |
| SE32: . . . you can usually reduce your abdominal pain by a **small amount** | 0.42 | 0.72 | 0.20 |
| SE33: . . . you can usually reduce **most** of your abdominal pain | 0.28 | 0.84 | 0.26 |
| SE34: . . . you can usually reduce your abdominal pain **completely** | 0.27 | 0.77 | 0.22 |

\**aya* refers to the women who worked in the study schools as janitors and caretakers. Schoolgirls typically had friendly relationships with these women, and if the school provided any menstrual pads for emergencies, it was typically the *aya* who could give one to a student.

recent menstrual period tended to have slightly higher scores on all three sub-scales than girls who only used other types of menstrual materials, although the differences were small and not statistically significant.

**Table 4. Correlations among oblimin oblique normalized rotated common factors.**

| Factors | Factor 1 (MHPM) | Factor 2 (MPM) | Factor 3 (EST) |
|---|---|---|---|
| Factor 1 (MHPM) | 1.00 | | |
| Factor 2 (MPM) | 0.41 | 1.00 | |
| Factor 3 (EST) | 0.50 | 0.24 | 1.00 |

Note: *MHPM* Menstrual hygiene preparation and maintenance, *MPM* Menstrual pain management, *EST* Executing stigmatized tasks

## Scale reliability

**Internal consistency.** Each of the three SAMNS-26 sub-scales showed good internal consistency in our sample. The MHPM sub-scale had a Cronbach's α of 0.86 and an average inter-item correlation of 0.26 (range from 0.26 to 0.27), the MPM sub-scale had an α of 0.87 and an average interitem correlation of 0.58 (range from 0.55 to 0.63), and the EST sub-scale had an α of 0.77 and an average interitem correlation of 0.46 (range from 0.37 to 0.52).

**Temporal stability.** The SAMNS-26 sub-scales showed good temporal stability after a mean interval of 15 days (Table 6). The concordance correlation coefficient for the MHPH sub-scale at time 1 and time 2 for the 42 girls who retested was 0.80 (95% CI = 0.69–0.91), for the MPM sub-scale 0.70 (95% CI = 0.56–0.85), and the EST sub-scale 0.71 (95% CI = 0.56–0.86). Bland-Altman plots indicated that differences between individuals' scores at time 1 and 2 tended to be smaller as average scale score increased, thus the measure had poorer temporal stability for those at the very lowest scale scores.

## Construct validity

SAMNS-26 sub-scores performed generally as expected in relation to validation measures (Table 7). The Beck Anxiety Inventory for Youth (BAI-Y) was negatively correlated with the MHPM sub-scale at -0.33, MPM at -0.17, and EST at -0.19. The Beck Self-Concept Inventory for Youth (BSCI-Y) was positively correlated with the MHPM sub-scale at 0.32, MPM at 0.19, and EST at 0.18. Our Bengali version of the Social Self-Efficacy Scale showed questionable internal consistency (α = 0.66) in our sample and was positively correlated with the SAMNS-26 sub-scales. The MHM sub-scale—which included the most items that required enlisting assistance from others—correlated the most strongly at 0.40. The EST sub-scale to a lesser degree contained items that required interaction with others, and it correlated with the Social Self-Efficacy Scale at 0.26. The MPM sub-scale correlated at 0.21. All correlations with validations measures were statistically significant with $p < 0.01$.

**Table 5. Self-Efficacy in Addressing Menstrual Needs scores by geography and menstrual materials used while away from home during last menstrual period (n = 381).**

| | Geography | | Menstrual material used while away from home during last period | |
|---|---|---|---|---|
| | Urban (n = 200) | Rural (n = 181) | Used disposable pad (n = 268) | Used other materials (n = 113) |
| SAMN Menstrual Hygiene Preparation and Maintenance | 76.0 (15.8) | 72.5 (17.2) | 74.6 (15.7) | 73.7 (18.5) |
| SAMN Menstrual Pain Management | 65.9 (26.0) | 63.3 (27.0) | 65.6 (25.4) | 62.3 (28.8) |
| SAMN Executing Stigmatized Tasks | 46.5 (30.0) | 44.4 (27.0) | 46.6 (28.3) | 42.9 (29.2) |
| **SAMNS Total** | **69.5 (16.6)** | **66.4 (16.8)** | **68.6 (16.0)** | **66.7 (18.4)** |

Note: Numbers are mean (SD)

(No differences between groups were significant at a level of $p <0.05$ based on Wilcoxon rank-sum tests)

**Table 6. Temporal stability of Self-Efficacy in Addressing Menstrual Needs Scale (SAMNS) scores after mean interval of 15 days (n = 42).**

| Sub-Scales | First testing | | Second testing | | Concordance Correlation Coefficient (95% CI) |
|---|---|---|---|---|---|
| | **M** | **SD** | **M** | **SD** | |
| Menstrual hygiene preparation and maintenance | 74.4 | 16.6 | 75.6 | 17.9 | 0.80 (0.69–0.91) |
| Menstrual pain management | 64.7 | 26.5 | 70.6 | 21.3 | 0.70 (0.56–0.85) |
| Executing stigmatized tasks | 45.5 | 28.6 | 54.8 | 28.3 | 0.71 (0.56–0.86) |
| **SAMNS Total** | **68.1** | **16.8** | **71.4** | **18.2** | **0.82 (0.72–0.92)** |

Table 8 displays SAMNS-26 sub-scale scores according to girls' responses to three validation items from the main study's endline survey. Wilcoxon rank-sum tests indicated that those who endorsed the item "During my last period, I felt anxious at school because of my menstruation," had lower scores across SAMNS-26 sub-scales than those who disagreed with the statement. Conversely, those who endorsed the item "During my last period, I felt comfortable at school," had higher scores across SAMNS-26 sub-scales than those who disagreed with the statement. Those who endorsed the item "I prefer to stay at home during my period," had lower scores across SAMNS-26 sub-scales than those who disagreed. All differences were significant at level $p < 0.01$ except for the differences in MPM sub-scale scores for each validation item, which were not statistically significant. SAMNS-26 sub-scale scores were not significantly correlated with months since menarche (MHPM: $\rho = 0.01$, $p = 0.80$; MPM: $\rho = -0.02$, $p = 0.71$; EST: $\rho = -0.07$, $p = 0.18$).

## Discussion

We developed and tested items for inclusion on the Self-Efficacy in Addressing Menstrual Needs Scale—a tool for assessing schoolgirls' confidence in their capabilities to accomplish tasks involved in addressing their menstrual needs. We developed an initial item pool grounded in a review of the existing literature and data from focus group discussions with schoolgirls. We sought input on our conceptualization of the construct and initial item pool from menstrual health experts and incorporated their feedback during item revision. An iterative process of cognitive interviewing, revising, and field piloting resulted in a 34-item scale questionnaire which we administered in a survey of 381 schoolgirls in Bangladesh. Through exploratory factor analysis and reliability and validity analyses, we reduced the questionnaire to a final 26-item version (SAMNS-26) comprising three intercorrelated sub-scales that each reliably measure distinct yet associated sub-domains of the larger SAMN construct: menstrual hygiene preparation and maintenance self-efficacy, self-efficacy in executing stigmatized tasks, and menstrual pain management self-efficacy.

The 17-item MHPM sub-scale includes items regarding self-efficacy in preparing for menstruation (e.g., tracking one's cycle and anticipating days of bleeding), accomplishing various tasks related to maintaining menstrual hygiene (e.g., obtaining, using, cleaning, and disposing

**Table 7. Correlations between Self-Efficacy in Addressing Menstrual Needs Scale (SAMNS) scores and validation measures (n = 381).**

| | Beck Anxiety Inventory-Youth | Beck Self-Concept Inventory-Youth | Social Self-Efficacy Scale |
|---|---|---|---|
| Menstrual hygiene preparation and maintenance | -0.33* | 0.32* | 0.40* |
| Menstrual pain management | -0.17* | 0.19* | 0.21* |
| Executing stigmatized tasks | -0.19* | 0.18* | 0.26* |
| SAMNS Total | -0.31* | 0.31* | 0.38* |

* $p < 0.01$

**Table 8. Mean Self-Efficacy in Addressing Menstrual Needs Scale (SAMNS) scores by responses to validation items (n = 381).**

| | Overall SAMNS Scores M (SD) | Menstrual Hygiene Prep & Maintenance Sub-Scale Scores M (SD) | Menstrual Pain Management Sub-Scale Scores M (SD) | Executing Stigmatized Tasks Sub-Scale Scores M (SD) |
|---|---|---|---|---|
| During my last period, I felt anxious at school because of my menstruation | | | | |
| Agree | 63.6 (17.4) | 69.8 (17.1) | 61.6 (28.4) | 40.1 (29.2) |
| Disagree | 70.9 (15.8) | 77.2 (15.6) | 66.6 (25.1) | 49.0 (27.7) |
| Wilcoxon rank-sum test | z = 3.8* | z = 4.2* | z = 1.5 | z = 3.0* |
| During my last period, I felt comfortable at school | | | | |
| Agree | 69.9 (16.3) | 76.1 (15.9) | 66.3 (26.0) | 48.3 (29.1) |
| Disagree | 63.4 (17.0) | 70.1 (17.5) | 60.6 (27.3) | 38.8 (26.2) |
| Wilcoxon rank-sum test | z = -3.2* | z = -3.1* | z = -1.9 | z = -2.8* |
| I prefer to stay at home during my period | | | | |
| Agree | 65.0 (16.4) | 71.1 (16.3) | 63.7 (27.2) | 40.3 (28.1) |
| Disagree | 71.9 (16.5) | 78.3 (16.1) | 65.9 (25.6) | 52.0 (28.0) |
| Wilcoxon rank-sum test | z = 4.2* | z = 4.7* | z = 0.66 | z = 3.9* |

*Difference significant at level $p < 0.01$

menstrual materials in a variety of contexts), and seeking assistance when needed. The MHPM sub-scale was most highly correlated with the 4-item EST sub-scale, which also included items related to obtaining and disposing menstrual materials and seeking assistance—but in situations where accomplishing the tasks involves greater risk of revealing one's menstrual status to a male person. The MHPM sub-scale was slightly less highly correlated with the MPM sub-scale which assesses girls' confidence in their capabilities to mitigate menstrual pain. The EST and MPM sub-scales were weakly correlated. This indicates that even if a girl becomes highly self-efficacious in executing stigmatized tasks to meet her menstrual hygiene needs, she could still have relatively low self-efficacy in mitigating her menstrual pain effectively.

We modeled our MPM items after those of Bandura's Pain Management Self-Efficacy scale [13], adapting wording to focus on the mitigation of dysmenorrhea specifically. We focused on abdominal pain because it was the most salient type of menstrual pain mentioned by girls during our qualitative research, and the physical complaint most commonly reported by schoolgirls in a previous study in Bangladesh [39]. However, we acknowledge that menstrual pain and discomfort comprise a host of additional symptoms (e.g., headache, leg pain, etc.), and could be heavily influenced by menstrual disorders such as endometriosis. Our study was not able to distinguish between girls experiencing primary and secondary dysmenorrhea and how this might affect their responses to the MPM sub-scale. Further research is warranted to more clearly define this sub-domain, identify additional items that could strengthen the MPM sub-scale's content validity, and explore what impact the experience of secondary dysmenorrhea has on how girls interpret and respond to items. Self-efficacy theory would suggest that changes in menstrual patterns may force girls to re-evaluate their self-efficacy beliefs as their routinized methods for addressing their needs become disrupted. This hypothesis is supported by a qualitative study of women with menstrual disorders in inner-city London which found that women who experienced increased heaviness or irregularity of their menses lost confidence in their previous strategies to "manage their menstruation" [40].

When researchers and program evaluators assess self-efficacy using single-item measures or tools that measure confidence as a generalized personality trait, they do not account for the

three dimensions of the self-efficacy construct as conceptualized by Bandura. We found it particularly important to account for the level/magnitude dimension of the construct by incorporating items that tap higher levels of difficulty in order to avoid creating a tool with a strong ceiling effect, as recommended by researchers who developed a self-efficacy scale for diabetes self-management in children [17]. They noted an early version of their diabetes self-management scale had a strong ceiling effect because the items reflected many of the basic activities necessary to manage diabetes that become routine to children who have managed their diabetes for a few years [17]. We anticipated that similarly, many of the basic tasks required to address one's menstrual needs could quickly become routine for a girl who has experienced many menstrual cycles. We therefore intentionally used the vignette activities in our focus group discussions to understand where various tasks generally fall along a continuum of difficulty for girls in our study context and thus enabled us to write sufficiently difficult items. As a result, we did not see a strong ceiling effect in the SAMNS-26. To ensure future iterations of the scale are appropriately targeted for other contexts, we recommend researchers conduct similar vignette activities with their study population to elicit lists of locally relevant tasks and their relative difficulty levels.

Our pretesting, piloting, and formal fielding of the scale questionnaire showed that girls in our study context were able to easily understand items, and it can be largely self-administered. However, girls in our study context had limited prior experience responding to questions using Likert-type response options, so survey enumerators provided careful verbal instructions. During early feasibility testing, girls who did not understand the instructions correctly the first time still provided responses to the test item rather than admitting they were unclear on what was being asked of them. Survey enumerators were only able to detect and correct miscomprehension by observing unreasonable responses to the test item. Bandura recommended administering a physical performance test to help familiarize child respondents with how to respond to self-efficacy scales [13]. To reduce measurement error, we likewise encourage researchers and program evaluators begin with a test of comprehension similar to the "pen test" we used to ensure girls understood the instructions before completing the questionnaire on their own [13]. Further research could explore ways of implementing an automated comprehension check (e.g., on a digital device) to enable fully self-administered and remote data collection.

Our study provides initial evidence in support of the construct validity of the SAMNS-26. Girls who scored higher on SAMNS-26 sub-scales reported lower feelings of anxiety and discomfort, in keeping with self-efficacy theory. Girls with higher SAMNS-26 sub-scale scores were also less likely to prefer to stay home during their menstrual periods. There was no significant correlation between the length of time since menarche and girls' self-efficacy. This may demonstrate that just as additional experience with menstruation provides girls opportunities to increase their self-efficacy through successful enactive experience and witnessing successful behavior modeling by similar others, it also provides opportunities to diminish self-efficacy. Experiencing what girls may consider to be "failures" in meeting their menstrual needs—such as experiencing visible blood stains on their clothing and being ridiculed for it—could have negative effects on girls' self-efficacy beliefs over time. This is consistent with the findings of another study (published after our data collection took place) that used a single item measure to assess girls' confidence to manage their menstrual period at school in Bangladesh and found no significant relationship between confidence and time since menarche [41].

The SAMNS-26 is designed for researchers and program implementers to include as part of needs assessment exercises prior to developing menstrual health programs or as a measure on pre- and post-intervention surveys to assess changes resulting from such programs. Incorporating self-efficacy measurement in menstrual health research can improve our understanding

of the pathways through which antecedent factors impact experiences of menstruation and subsequent outcomes related to health, wellbeing, and social participation [1]. We developed the scale items based on our qualitative research that identified tasks schoolgirls in Bangladesh carry out in meeting their menstrual needs; it was not our intention to develop a universal measure for comparing schoolgirls' self-efficacy beliefs across global contexts. Doing so would sacrifice the scale's explanatory and predictive power in the Bangladeshi context and thus be less useful in guiding program development and evaluation locally.

However, the tool is general enough that it can be completed by girls regardless of their preferred type of menstrual material (e.g., pad, cloth, etc.) within the Bangladeshi context and across both urban and rural communities. It can also be used as a model for structuring items, response options, and instructions by those wishing to measure adolescent schoolgirls' self-efficacy in addressing their menstrual needs in contexts other than Bangladesh. This should be accompanied by iterative pre-testing to ensure the items are locally relevant and easily interpretable in the new setting. Our cognitive interview process can be used as an example approach to pre-testing. Items that are extremely specific to our context may need to be dropped or adapted before administering the tool in other settings. For example, our scale contains an item referencing obtaining help from an *aya*—a female school staff member (with janitorial and other support roles) who could unlock latrines and provide menstrual pads upon request from students. A staff member with a similar role may not exist in other settings, but the item might be adapted to refer to another trusted member of staff.

## Limitations

Ideally our draft scale questionnaire would have contained 2–3 times the number of items we retained in our final validated scale [35, 42]. However, we had to restrict our pool of potential items to 34 due to time constraints of the larger survey on which this scale questionnaire was tested. Resultantly, it is possible that we may not have enough items to fully tap all sub-domains [43] and the SAMNS construct could comprise more than the three sub-domains identified in our exploratory factor analysis. However, the rigor of the item development and pretesting process may have translated to fewer, yet better performing, items. Further testing of additional items across categories of menstrual care tasks (particularly regarding symptom management) as well as confirmatory factor analyses in a new, larger sample is recommended to determine if the factor structure remains consistent. Tests for measurement invariance should also be performed [44] to verify whether the construct can be measured with the same model across important sub-groups of schoolgirls, such as those with and without secondary dysmenorrhea or menorrhagia. We also would have liked to test the psychometric properties of the SAMNS-26 in a sample of schoolgirls with large variations in their physical and economic environments. However, we were limited to only including participants who had recently received a 6-month intervention intended to create more supportive school environments for menstruating girls. This may have resulted in reduced response variation across items being tested given the more supportive than average school environment, which could have led to some items being dropped from the final scale that would have performed well in other samples. The SAMNS-26 was developed with schoolgirls, and therefore the perspectives of girls who have dropped out or were never enrolled in school are not reflected in the tool.

## Conclusion

We developed a 26-item scale comprising three sub-scales that measure schoolgirls' self-efficacy in addressing their menstrual needs. In our testing sample of 381 girls attending rural and urban schools in Bangladesh, the scale demonstrated favorable reliability and construct

validity. The SAMNS-26 can be used for further intervention research and menstrual health program evaluations with schoolgirls in Bangladesh from the time of menarche up to Class 10 (age 17), providing a way to assess changes in self-efficacy beliefs over time. The tool enables incorporation of self-efficacy into multivariate models for exploring the relationships among antecedents to menstrual experiences and hypothesized impacts on health, wellbeing, and education attainment. Additional testing of the tool in new samples in Bangladesh and other contexts globally is recommended to strengthen evidence of its validity. We recommend researchers and program evaluators take a similar iterative approach to pretesting scale items when adapting and revalidating the tool for new contexts.

## Supporting information

**S1 Fig. Integration of a self-efficacy sub-study to develop and validate the Self-Efficacy in Addressing Menstrual Needs Scale within the main study 'piloting MHM interventions among urban and rural schools in Bangladesh', 2017–2018.**
(PDF)

**S2 Fig. Example vignette activity as part of focus group discussions with schoolgirls during the development of the Self-Efficacy in Addressing Menstrual Needs Scale in Bangladesh, 2017–2018.**
(PDF)

**S1 Table. Socio-demographic information of focus group discussion and cognitive interview participants during the development of the Self-Efficacy in Addressing Menstrual Needs Scale in Bangladesh, 2017–2018.**
(PDF)

**S2 Table. Exemplar tasks across categories which girls enact to address their menstrual needs, as reported by schoolgirls in Bangladesh during focus group discussions for the development of the Self-Efficacy in Addressing Menstrual Needs Scale, 2017–2018.**
(PDF)

**S3 Table. Characteristics of post-menarcheal schoolgirls who participated in a survey to test items for the development of the Self-Efficacy in Addressing Menstrual Needs Scale in Bangladesh, 2018.**
(PDF)

**S4 Table. Item response mean, standard deviation, skew, and kurtosis for 34 items formally tested with schoolgirls (n = 381) for the development of the Self-Efficacy in Addressing Menstrual Needs Scale in Bangladesh, 2018.**
(PDF)

**S5 Table. Items dropped during psychometric analyses of responses from 381 post-menarcheal schoolgirls in Bangladesh during the testing of the Self-Efficacy in Addressing Menstrual Needs Scale, 2018.**
(PDF)

**S1 File. The self-efficacy in addressing menstrual needs section of the self-efficacy sub-study survey (34 items for formal testing).**
(PDF)

**S2 File. Description of survey measures included in the testing of the Self-Efficacy in Addressing Menstrual Needs Scale in Bangladesh, 2018.**
(PDF)

**S3 File. Self-Efficacy in Addressing Menstrual Needs Scale (SAMNS-26) [English translation of Bengali version].**
(PDF)

**S4 File. Self-Efficacy in Addressing Menstrual Needs Scale (SAMNS-26) [Bengali version].**
(PDF)

**S5 File. Inclusivity questionnaire.**
(DOCX)

# Acknowledgments

We thank the girls who gave their time to participate in the study and the school staff who enabled us to work in their schools. We acknowledge our icddr,b team members who contributed to portions of data collection: Shifat Khan, Farhana Akand Kona, Shirina Akter Shilpi, Dr. Farjana Jahan, Shaan Muberra Khan, and Dr. Mehjabin Tishan Mahfuz. We acknowledge Maxim Argho for his translation support in the broader study and Muhammad Kamal Uddin, Professor and Chairman of the Department of Psychology at the University of Dhaka for his *Bangla* version of the Beck Youth Inventory. We acknowledge Associate Professor Caitlin Kennedy of the Johns Hopkins Bloomberg School of Public Health for her advisement to the first author and the menstrual health experts who provided feedback on the draft items.

# Author Contributions

**Conceptualization:** Erin C. Hunter.

**Data curation:** Erin C. Hunter, Mahbub-Ul Alam, Supta Sarker, Moshammot Mobashara, Marufa Momata.

**Formal analysis:** Erin C. Hunter, Nazrin Akter.

**Funding acquisition:** Erin C. Hunter, Farhana Sultana, Peter J. Winch.

**Investigation:** Erin C. Hunter, Supta Sarker, Nazrin Akter, Moshammot Mobashara, Marufa Momata.

**Methodology:** Erin C. Hunter, Sarah M. Murray, Peter J. Winch.

**Project administration:** Erin C. Hunter, Farhana Sultana, Supta Sarker, Nazrin Akter, Moshammot Mobashara.

**Resources:** Mahbubur Rahman.

**Software:** Erin C. Hunter, Mahbub-Ul Alam, Supta Sarker.

**Supervision:** Erin C. Hunter, Sarah M. Murray, Farhana Sultana, Mahbubur Rahman, Peter J. Winch.

**Visualization:** Erin C. Hunter.

**Writing – original draft:** Erin C. Hunter.

**Writing – review & editing:** Erin C. Hunter, Sarah M. Murray, Farhana Sultana, Mahbub-Ul Alam, Supta Sarker, Mahbubur Rahman, Peter J. Winch.

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
