## [Decision Letter · Decision Letter 0]

28 Jun 2022

PONE-D-21-37712Development and validation of the Self-Efficacy in Addressing Menstrual Needs Scale (SAMNS-26) in Bangladeshi Schools: A measure of girls' menstrual care confidencePLOS ONE

Dear Dr. Hunter,

Thank you for submitting your manuscript to PLOS ONE. After careful consideration, we feel that it has merit but does not fully meet PLOS ONE’s publication criteria as it currently stands. Therefore, we invite you to submit a revised version of the manuscript that addresses the points raised during the review process. Both reviewers are complimentary of the paper and have some comments throughout the manuscript to address.   The attachment provided by reviewer 1 also suggests some improvements to the discussion.

We look forward to receiving your revised manuscript.

Kind regards,

Alison Parker

Academic Editor

PLOS ONE

6. Please note that in order to use the direct billing option the corresponding author must be affiliated with the chosen institute. Please either amend your manuscript to change the affiliation or corresponding author, or email us at plosone@plos.org with a request to remove this option.

Reviewers' comments:

Reviewer's Responses to Questions

**Comments to the Author**

1. Is the manuscript technically sound, and do the data support the conclusions?

Reviewer #1: Yes

Reviewer #2: Yes

2. Has the statistical analysis been performed appropriately and rigorously? 

Reviewer #1: Yes

Reviewer #2: Yes

3. Have the authors made all data underlying the findings in their manuscript fully available?

Reviewer #1: No

Reviewer #2: Yes

4. Is the manuscript presented in an intelligible fashion and written in standard English?

Reviewer #1: Yes

Reviewer #2: Yes

5. Review Comments to the Author

Reviewer #1: Please note that question 3 has a yes / no answer only, and as such I have had to put 'no'. The authors have provided considerable information within text and as supplementary information. However, they have not provided the audio data - with reasons given (and in my opinion appropriately so, particularly as the supplementary file provided is sufficient). They also have not provided quantitative data but will do so after acceptance of the manuscript. (Please note I have stated that the statistical analysis has been performed appropriately and rigorously - a yes / no response required - based on the authors detailed description of the steps taken, and their justification of each, along with the details within the text and supplementary information)

Reviewer #2: Rigor in gathering data on menstrual health and hygiene measurements has been limited to date. Lack of accurate data will severely hamper progress on improving MHH for girls and women.

Self-efficacy in addressing menstrual needs has been particularly overlooked, and this article is thus an important contribution to a very limited literature and will help inform both researchers and programmers how to ask and utilise these data.

The researchers have taken rigorous steps in developing and validating their measures through on the ground lived experiences of school girls, ensuring that the language used is appropriate; Refinement of the tool by seeking external ‘menstrual experts’ also facilitates generalisability of the tool. Their methodology of testing and retesting, followed by use of factor analysis to assess internal consistency, temporal stability, and construct validity further enhances the strength of this work.

Only a few small suggestions, as below, but overall an excellent contribution to the literature.

Creation of pool of items based on FGD of schoolgirls in rural and urban Bangladesh … ages ? any divergent opinions between younger and older girls – how was this resolved when creating te tool?

Refinement using 21 cognitive interviews – not the same girls as the FGD , can authors elaborate how able to select same diverse participants among 21 ? by age, by age menarche, by rural v urban ?

Were standard metrics for anxiety e.g. PHQ-9 used to test validity against menstrual item? If not why?

Pilot tested in 13 girls in urban school only .. any bias by not including rural (?less educated)

Boys suddenly creep in in the psychometric testing – but not involved previously – how relevant are their responses ? in results 404 girls participating are mentioned but not the boys. Boys testing of the tool was not really mentioned in the discussion either – anything to add?

Doesn’t say what the 4 intervention schools for psychometric testing are - were they 2 rural and 2 urban, if only urban – why and would this have any bearing on final outcomes?

6. PLOS authors have the option to publish the peer review history of their article (what does this mean?). If published, this will include your full peer review and any attached files.

Reviewer #1: **Yes: **Linda Mason

Reviewer #2: No

---

## [Author Response · Author response to Decision Letter 0]

30 Aug 2022

Dear Reviewers,

Thank you for your review of our manuscript and the helpful suggestions for strengthening the paper. We have itemized our responses to the comments below.

Response to Editor’s comments:

1. Comment: Please ensure that your manuscript meets PLOS ONE's style requirements, including those for file naming. The PLOS ONE style templates can be found at

Response: We have edited the title of the manuscript to remove unnecessary capitalization and bring the manuscript into alignment with the style requirements. 

2. Comment: Please include a complete copy of PLOS’ questionnaire on inclusivity in global research in your revised manuscript. Our policy for research in this area aims to improve transparency in the reporting of research performed outside of researchers’ own country or community. The policy applies to researchers who have travelled to a different country to conduct research, research with Indigenous populations or their lands, and research on cultural artefacts. The questionnaire can also be requested at the journal’s discretion for any other submissions, even if these conditions are not met. Please find more information on the policy and a link to download a blank copy of the questionnaire here: https://journals.plos.org/plosone/s/best-practices-in-research-reporting. Please upload a completed version of your questionnaire as Supporting Information when you resubmit your manuscript.

Response: We have included the completed questionnaire with our resubmission.

3. Comment: We note that the grant information you provided in the ‘Funding Information’ and ‘Financial Disclosure’ sections do not match. When you resubmit, please ensure that you provide the correct grant numbers for the awards you received for your study in the ‘Funding Information’ section.

Response (revised Financial Disclosures statement to be published with the manuscript): The 'Piloting MHM interventions among urban and rural schools in Bangladesh' study (the “main study”) was funded by the Bill and Melinda Gates Foundation (OPP1140650) (https://www.gatesfoundation.org/) to FS. icddr,b acknowledges with gratitude the commitment of BMGF to its research efforts. icddr,b is also grateful to the Governments of Bangladesh, Canada, Sweden, and the UK for providing core/unrestricted support. Additional support for data collection towards the self-efficacy sub-study was provided through a Dissertation Enhancement Award to ECH from the Center for Qualitative Studies in Health and Medicine of Johns Hopkins Bloomberg School of Public Health (https://www.jhsph.edu/departments/health-behavior-and-society/research-and-centers/center-for-qualitative-studies-in-health-and-medicine/). The funders had no role in study design, data collection and analysis, decision to publish, or preparation of the manuscript.

4. Comment: In your Data Availability statement, you have not specified where the minimal data set underlying the results described in your manuscript can be found. PLOS defines a study's minimal data set as the underlying data used to reach the conclusions drawn in the manuscript and any additional data required to replicate the reported study findings in their entirety. All PLOS journals require that the minimal data set be made fully available. For more information about our data policy, please see http://journals.plos.org/plosone/s/data-availability.

Response: To align with the informed consent provided by study participants, approval is needed for other researchers to access the quantitative data. Data are available from the icddr,b institutional data repository for researchers upon approval of a Data Licensing Application & Agreement. For more information, see https://www.icddrb.org/component/content/article/10003-datapolicies/1893-data-policies. Request for icddr,b research data should be addressed to Ms. Armana Ahmed, Head, Research Administration at aahmed@icddrb.org. Making the qualitative data (audio recordings) publicly available would compromise the confidentiality we promised to study participants, as girls could be identifiable by their voices and stories. We have provided a high-level summary of qualitative data generated during focus group discussions as supporting info with the published paper.

5. Comment: Your ethics statement should only appear in the Methods section of your manuscript. If your ethics statement is written in any section besides the Methods, please move it to the Methods section and delete it from any other section. Please ensure that your ethics statement is included in your manuscript, as the ethics statement entered into the online submission form will not be published alongside your manuscript.

Response: We have removed a standalone section on Ethical Approval from the manuscript and it is now part of the Methods section.

6. Comment: Please note that in order to use the direct billing option the corresponding author must be affiliated with the chosen institute. Please either amend your manuscript to change the affiliation or corresponding author, or email us at plosone@plos.org with a request to remove this option.

Response: The study was funded by the Bill and Melinda Gates Foundation (Grant Number OPP1140650), which requires the findings to be published open access. The Foundation requires that journals send the invoice to the Foundation so they can pay on behalf of the research team. The Foundation requires that I put the grant number in the acknowledgements section of the manuscript, but the PLOS ONE guidelines disallowed this, so I am unsure how to meet the requirements of both the funder and the journal. I have provided information below for invoicing:

Invoices for publishing fees must be addressed to the Gates Foundation & emailed to openaccess@gatesfoundation.org.

Bill & Melinda Gates Foundation c/o Open Access

PO Box 23350

Seattle, WA 98102-0650

United States

Phone: +1 206-709-3278

VAT does not apply & the foundation is not tax-exempt for this purpose

Invoices must also include:

• Gates Grant Number in the Purchase Order Field

• Article DOI or Title

• Indication that the article license is CC-BY

Responses to Reviewers' comments:

1. Comment: Rigor in gathering data on menstrual health and hygiene measurements has been limited to date. Lack of accurate data will severely hamper progress on improving MHH for girls and women. Self-efficacy in addressing menstrual needs has been particularly overlooked, and this article is thus an important contribution to a very limited literature and will help inform both researchers and programmers how to ask and utilise these data. The researchers have taken rigorous steps in developing and validating their measures through on the ground lived experiences of school girls, ensuring that the language used is appropriate; Refinement of the tool by seeking external ‘menstrual experts’ also facilitates generalisability of the tool. Their methodology of testing and retesting, followed by use of factor analysis to assess internal consistency, temporal stability, and construct validity further enhances the strength of this work. Only a few small suggestions, as below, but overall an excellent contribution to the literature.

Response: We thank you for your detailed review of our paper and appreciate your comments that the work will be an excellent contribution to the literature. We have responded to each of your recommendations below.

2. Comment: Creation of pool of items based on FGD of schoolgirls in rural and urban Bangladesh … ages ? any divergent opinions between younger and older girls – how was this resolved when creating te tool?

Response: In the first line of the “Item pool development” section of the Results, we report that girls ages 11-16 years participated in the FGDs, and further information about the number of participants by age and “time since menarche” is presented in S1 Table. Because of our guiding theoretical framework, we were attuned to “time since menarche” as likely to be more influential on girl’s responses compared to simply age alone, since a 14-year-old girl who only reached menarche within the last few months would have much less experience managing her menstrual needs than a 14-year-old girl who had been menstruation for 3 years already. We aimed to create a pool of items that included a mix of difficulty levels--including some that would be relatively “easy” to endorse, and some that would be considered more “difficult” and some in between. So, we did not need to “resolve” divergent opinions between girls with varying levels of experience with menstruation, rather we purposefully wrote items based on the full range.

3. Comment: Refinement using 21 cognitive interviews – not the same girls as the FGD, can authors elaborate how able to select same diverse participants among 21? by age, by age menarche, by rural v urban ?

Response: To clarify this, we have added text briefly describing how we approached recruitment and selection of participants for our FGDs and cognitive interviews:

“To recruit and select participants, we worked with female schoolteachers to explain the purpose of the research in classrooms of female students and compile a list of post-menarcheal girls by class level and approximate time since menarche. Schoolgirls voluntarily self-identified their eligibility in the absence of males. We then instructed eligible participants on how to request written consent from parents to participate in the study and requested students return signed consent forms to us the following day. We later purposively selected participants from this master list of eligible girls for each data collection episode according to class level and length of time since menarche. Focus group discussions were convened with privacy in empty classrooms…” [and in the section on cognitive interviews:] “We purposively selected girls from the master list of eligible students who had not previously participated in FGDs and according to time since menarche to ensure a range of familiarity with menstrual care.”

4. Comment: Were standard metrics for anxiety e.g. PHQ-9 used to test validity against menstrual item? If not why?

Response: Yes, in the “Survey Measures” section of the Results section (with more information provided in a supplementary file), we report using the Beck Anxiety Inventory (Youth) for validity testing. This tool had been previously validated with adolescents in Bangladesh.

5. Comment: Pilot tested in 13 girls in urban school only ... Any bias by not including rural (less educated?)

Response: Please see our response on this issue to the other Reviewer’s comments at the end of this document (Comment #19). Due to logistical and time constraints, the final field piloting was done using a convenience sample of girls in one of the urban schools as it was located a reasonable distance from icddr,b’s office and study team members were already visiting the school those days for other project activities. Our prior intensive cognitive interviews had already enabled us to finalize items to a point where we were reasonably confident they would be comprehensible to students across ages and geography. As our primary intent for the final field piloting was to generate a rough estimate of how long it might take girls to complete the item pool on their own and thus if the formal testing on the main study’s endline survey would likely be feasible (which it ultimately was), we do not feel that sampling from only the urban school for this portion biases the tool or the conclusions of the study.

6. Comment: Boys suddenly creep in in the psychometric testing – but not involved previously – how relevant are their responses? in results 404 girls participating are mentioned but not the boys. Boys testing of the tool was not really mentioned in the discussion either – anything to add?

Response: Thank you for the opportunity to clarify. Boys participated in the main study’s endline survey. However, only girls who indicated on the main survey that they had experienced menstruation then went on to complete the self-efficacy sub-study survey. To reduce confusion further, we have decided to remove any mention of boys from the manuscript, as they were not part of the self-efficacy sub study even though they participated in the main study activities and intervention.

7. Comment: Doesn’t say what the 4 intervention schools for psychometric testing are - were they 2 rural and 2 urban, if only urban – why and would this have any bearing on final outcomes?

Response: The psychometric testing was done in two rural and two urban schools. In the first paragraph of the “Study Setting” section of the Methods, we state that “The main study comprised a formative research phase in four of the schools (two urban, two rural) to inform the development of an intervention package, followed by a six-month piloting period in four other schools (two urban, two rural) to evaluate the intervention…”. Additionally, in the final sentence of the “Overview of research design” section of the Methods, we say that “In Phase 4, we assessed the psychometric properties of the tool through testing items on a survey of randomly selected schoolgirls in urban and rural Bangladesh.” We had refrained from repeating this again in later sections, but now we have clarified this further by adding a parenthetical note in the section “Phase 4: Psychometric testing” so it now reads: “In April 2018, we leveraged the main study’s endline survey in the four intervention schools (two urban, two rural) to collect data for assessing…”.

8. Comment: The paper describes the development of a new tool to assess psychosocial components of menstruation as experienced by adolescent schoolgirls. Evidence to date suggests that poor menstrual hygiene impacts negatively on girls education leading to school absence and drop-out, girls who remain in school whilst menstruating face issues such as inability to concentrate on their lessons, and period shaming. The negative impact on girls’ education contributes towards gender inequality. The tool described in this paper has been developed to assess the psychosocial components of schoolgirls menstruation and can therefore be used to develop interventions improving the experience of menses. It is likely that this could positively impact on girls education and go some way to help address gender imbalance in the future. I would like to commend the authors on this well written paper. The development of the tool was very thorough. There is much detail provided in the paper and each step was very clearly described. This left me with few questions or sections where I needed further clarification.

Response: We thank you for your comments and appreciate your thorough review of our paper.

9. Comment: One concern that I have is that whilst the authors are realistic and state that the tool was developed and validated only in the Bangladeshi context, and not intended to be a universal measure, this does weaken its relevance to PLOS One readers generally. I feel that the discussion is relatively weak compared to the rest of the paper, and one way of enhancing might be to provide more clarification on how the tool could be amended for wider use – without following all of the same procedures as described in this paper, which clearly involved much detailed fieldwork, in order to recreate a similar model. I appreciate that there are other issues involved in menstrual care that might be relevant to other contexts, but from my own experience in HIC and LIC settings the list of items in SAMNS-26 seems applicable generally, (bar the one item specific to aya, as highlighted by the authors). I appreciate that to some extent lines 604 – 625 do touch on this, as does line 656-657. Perhaps just to turn around the narrative a bit in order to be more positive and definitive about the use of the SAMNS-26 and how it could be taken forward or adapted, might open this out more to a wider audience. For example, perhaps line 655 could read along the lines of ‘Additional testing of the tool in the Bangladeshi and similar contexts is recommended…..’. 

Response: Thank you for this helpful recommendation. We have taken this into account and modified our phrasing in the Discussion and Conclusion sections to take this more positive tone and recommend how it might be adapted in other contexts.

10. Comment: In addition to the above suggestion, I would like to see the discussion strengthened to do justice to the work that has been undertaken. In particular, the precis of the methods in the first couple of paragraphs is rather repetitive. This could be removed and replaced with a succinct statement of what this study has achieved with room then for more in-depth discussion of strengths and future use of the tool (and areas for development) incorporated instead.

Response: We have modified the Discussion section to take this into account. However, we have left the precis at the beginning of the section to aid readers less familiar with scale development methodologies recall how each of the phases fit together to yield the final tool.

11. Comment: I also would like to see the discussion incorporating more literature. Currently, it references just 3 studies but would benefit from greater alignment with a wider body of literature to confirm or refute study methods and / or findings and ways forward to assist management of menstrual hygiene. It might also be interesting to discuss the SAMNS-26 scores – I was particularly interested to note there were only small differences in score between those using disposable pads compared to other materials when away from home, (also that sub-scale scores, particularly Preparation and Task were not correlated with months since menarche. This is picked up on in the discussion but no references are given to assist the reader in understanding why you may have developed your hypothesis, nor to evidence your (reasonable) suggestion that time provides opportunity to diminish self-efficacy). These would strengthen your arguments made.

Response: We have now provided additional references at multiple points in the Discussion section to more clearly indicate the alignment of our methods and findings with a wider body of literature—including the point you have highlighted about scores not being correlated with months since menarche.

12. Comment: The introduction makes mention that individuals with low self-efficacy may (theoretically) avoid challenging situations such as attending school during menstruation. As repeated absence is linked with school drop-out would it be useful to mention perhaps as a limitation of the study, that it is school-based only and therefore may miss those girls who experience the greatest levels of menstrual related stress and anxiety?

Response: We have added this to the limitations section: “The SAMNS-26 was developed with schoolgirls, and therefore the perspectives of girls who dropped out or were never enrolled in school are not reflected in the tool.”

13. Comment: I have little specific feedback, because as stated, I feel that the authors have written a paper that is of a good standard.

Response: Thank you.

Abstract

14. Comment: Line 36 – add a few words to incorporate the importance of measuring psychosocial components or why it is needed

Response: We have added the following to the beginning of the abstract to demonstrate the importance of enabling measurement of a construct that has been identified in the qualitative literature as an important component of menstrual experiences that may impact health, education, and social participation: “Qualitative studies have described girls’ varying levels of confidence in managing their menstruation, with greater confidence hypothesized to positively impact health, education, and social participation outcomes. Yet, measurement of this and other psychosocial components of adolescent girls’ menstrual experiences has been weak…”

15. Comment: Line 62- the abstract conclusion suggests further research should explore whether secondary dysmenorrhoea impacts how girls interpret and respond to items. However, this is not mentioned as a conclusion in the main paper, although it is mentioned in the discussion. I suggest that it is removed from here, or if the authors think it is a key take home message, then it should be in the actual study conclusion.

Response: We agree with your suggestion and have now edited the final sentence in the abstract to more closely match the conclusions in the main text: “Further testing of the tool is recommended to strengthen evidence of its validity in additional contexts.” 

Methods

16. Comment: Lines 138-153. I cannot easily reconcile the description provided (lines 138-145), nor the description of phase 1 (lines 148-153) with the diagram referred to as S1-fig. Phase 1 on the diagram refers to FGDs, in text it states ’we designed the questionnaire format and created an initial pool of draft items’. Could this be made clearer for the reader on the diagram please.

Response: S1 Fig is a diagram that only shows the integration of the self-efficacy sub-study activities into the broader main study. Our intention for this figure is to simply demonstrate how we leveraged some of the main study’s data collection events and then built-in additional data collection methods alongside the on-going main study in order to develop and validate the SAMNS-26. This is why we only reference the S1 Fig within the sentence of the manuscript that says: “The work described in this paper was a self-efficacy sub-study commenced halfway through the main study, after the formative research phase but immediately prior to implementation of a baseline survey in intervention schools (S1 Fig).”

In contrast, Fig 1 is intended to demonstrate what we describe in the overview of the research design section of the text (our four-stage process model for developing the scale). To clarify Fig 1 further, we have amended the bottom box in the Phase 1 portion of the diagram (added text shown in bold here): “9 FGDs with schoolgirls to identify tasks involved in addressing menstrual needs at varying levels of difficulty across multiple categories to enable drafting of initial item pool”

17. Comment: Lines 207-210. Please clarify if the ‘study team’ were the same as the ‘study enumerator team’ as described earlier. Do they include the authors? I ask because the FGDs were conducted in Bengali, no mention was made of translating. Were the audio recordings transcribed? It is not clear to me whether the analysis was done immediately as per lines 209 – 212 and as per the debrief.

Response: Thank you for the opportunity to clarify. The analysis of the FGD data was done immediately, as per the description in the manuscript (i.e., during intensive debriefing meetings immediately after FGD sessions).

The audio recordings were not transcribed/translated for the analyses presented in this paper (they were at a later date for additional analyses not presented in this paper). We audio recorded the FGDs and cognitive interviews primarily so that we would be able to refer to them if necessary (for instance, if we missed something in our notes and needed to double-check). The analyses were done immediately after the FGDs and cognitive interviews during intensive debriefing meetings among the study team members who were involved in the data collection event and facilitated by the first author. Debriefing meetings were conducted in a mix of Bengali and English. In effort to clarify this in the manuscript, we have added the following text:

“FGDs were audio recorded so we could check any gaps in field notes, but full transcriptions were not produced for the analyses presented in this paper.”

And

“Interviews typically lasted 1-1.5 hours and were audio recorded for verification purposes but not transcribed.”

The “14 female professional survey enumerators” (mentioned in the section where we describe the very early piloting of 10 test/dummy items within the main study’s baseline survey) had been already hired separately to implement the main study’s baseline survey. Since the main study’s baseline survey implementation was happening at the very beginning of the scale development work, we were not able to include a full self-efficacy scale questionnaire at that time, but we decided to leverage the opportunity to at minimum assess the feasibility of the planned format, instructions, and response options that we would use for the self-efficacy questionnaire. Thus, we added a session to the enumerator team’s existing training for the main study’s baseline survey to also include how to carry out the feasibility testing for the self-efficacy component when fielding the baseline survey. Authors MA and SS led the training of the data enumeration team for the main study’s baseline survey, and then author EH joined them for the portion of the training on the self-efficacy component.

Throughout the manuscript, the term “study team” refers to the group of researchers who carried out the self-efficacy sub-study, and so it was the “study team” that conducted all the rest of the data collection and analysis presented in this manuscript (e.g. FGDs, cognitive interviews, the survey for psychometric testing, etc.). Different members of the study team were engaged in different activities over the life of the sub-study, with the first author being across all of them. The study team comprised the authors (whose specific contributions are documented in the author contributions section of this publication) plus additional members of the “main study” team recognized in the acknowledgements section for their contributions to particular data collection episodes when their schedules allowed but did not meet full authorship criteria for the self-efficacy sub-study. 

18. Comment: Line 231 – why 21 cognitive interviews? Why 2-6 interviews per round? Were they conducted in Bengali? Were the interviews transcribed / translated given that they were audio recorded? I realise that the paper contains much detail but this did leave me wondering.

Response: We hope our response to point 17 above helps with most of these questions. The cognitive interviews were conducted in Bengali and were not transcribed in full. The audio recordings enabled us to refer back if necessary, but we took detailed field notes during the interviews and intensively debriefed immediately after conducting each of them during which we expanded our notes further and actively made analytical decisions together. We have clarified this in the text by adding the following sentence: “Interviews were conducted in Bengali, typically lasted 1-1.5 hours, and were audio recorded for verification purposes but not transcribed.” In regard to your question about the rationale for our sample, this is alignment with the literature concerning appropriate sample sizes and multiple rounds for cognitive testing studies (see for example, “The devil is in the detail…” by Scott et al. https://pubmed.ncbi.nlm.nih.gov/33978729/). The iterative approach enabled us to identify and address issues with items and test revised versions on the next round so that we did not needlessly test known problematic items with a large number of participants. We continued with the iterative revisions and testing until overall we achieved good match between our measurement intent for the items and participants’ comprehension. We have added text to the manuscript to more explicitly state this: “This iterative approach enabled us to identify and address issues with items and test revised versions until we achieved a good match between each item’s intent and participants’ interpretations (Scott, Ummer, & LeFevre, 2021).”

19. Comment: Line 252 – why 13 girls? How selected? Why 2 rounds?

Response: Similar to our approach to the intensive cognitive interviews, conducting the field piloting in multiple rounds provided an opportunity to address any issues before the following iteration. Our prior cognitive interviews had enabled us to finalize items to a point where we were reasonably confident they would be comprehensible to students, so our primary intention for the field pilot was more simply to gain an understanding of roughly how long it would take girls to complete the entire item pool (without the concurrent probing that took place during the earlier cognitive interviews). This was achievable with just 13 participants. We also used the opportunity to explore whether girls could respond on their own without assistance. They were able to, and our retrospective probing showed that girls’ interpretations matched our intentions for the items, so we did not conduct further rounds beyond two. Due to timeline and logistical constraints, the girls who participated in the final field piloting were selected by convenience from their classes. We have revised the manuscript to reflect this information: “We conducted a final field pilot in two rounds in April 2018 with a total of 13 girls selected by convenience from Classes 8-10 in an urban school to determine how long it took them to complete the revised pool of 35 items and if they could do so without assistance. The iterative approach provided an opportunity to make adjustments between rounds if necessary...”

---

## [Decision Letter · Decision Letter 1]

22 Sep 2022

Development and validation of the Self-Efficacy in Addressing Menstrual Needs Scale (SAMNS-26) in Bangladeshi schools: a measure of girls' menstrual care confidence

PONE-D-21-37712R1

Dear Dr. Hunter,

We’re pleased to inform you that your manuscript has been judged scientifically suitable for publication and will be formally accepted for publication once it meets all outstanding technical requirements.

Kind regards,

Alison Parker

Academic Editor

PLOS ONE

Additional Editor Comments (optional):

Reviewers' comments:

Reviewer's Responses to Questions

**Comments to the Author**

1. If the authors have adequately addressed your comments raised in a previous round of review and you feel that this manuscript is now acceptable for publication, you may indicate that here to bypass the “Comments to the Author” section, enter your conflict of interest statement in the “Confidential to Editor” section, and submit your "Accept" recommendation.

Reviewer #1: All comments have been addressed

Reviewer #2: All comments have been addressed

2. Is the manuscript technically sound, and do the data support the conclusions?

Reviewer #1: Yes

Reviewer #2: Yes

3. Has the statistical analysis been performed appropriately and rigorously? 

Reviewer #1: Yes

Reviewer #2: Yes

4. Have the authors made all data underlying the findings in their manuscript fully available?

Reviewer #1: No

Reviewer #2: Yes

5. Is the manuscript presented in an intelligible fashion and written in standard English?

Reviewer #1: Yes

Reviewer #2: Yes

6. Review Comments to the Author

Reviewer #1: I would like to congratulate the authors on the high standard of their important contribution to menstrual health research. The revisions made to my suggestions have been clarified / amended as appropriate - and I would also like to mention that whilst they made amendments to reflect how their study could be used by others working in the same field, so as to widen the target audience for this paper, they were realistic and did not overstate. This was very well done. Thank you for the opportunity to review this paper.

Reviewer #2: Al required questions have been answered well by the authors, and no other questions on revised manuscript

7. PLOS authors have the option to publish the peer review history of their article (what does this mean?). If published, this will include your full peer review and any attached files.

Reviewer #1: **Yes: **Linda Mason

Reviewer #2: No

---

## [Editor Report · Acceptance letter]

27 Sep 2022

PONE-D-21-37712R1 

Development and validation of the Self-Efficacy in Addressing Menstrual Needs Scale (SAMNS-26) in Bangladeshi schools: a measure of girls’ menstrual care confidence 

Dear Dr. Hunter:

I'm pleased to inform you that your manuscript has been deemed suitable for publication in PLOS ONE. Congratulations! Your manuscript is now with our production department. 

Kind regards, 

on behalf of

Dr. Alison Parker 

Academic Editor

PLOS ONE